# Observation on the Droplet Ranging from 2 to 16 μm in Cloud Droplet Size Distribution Based on Digital Holography

**Pan Gao, Jun Wang \*, Yangzi Gao, Jingjing Liu and Dengxin Hua**

School of Mechanical and Precision Instrument Engineering, Xi'an University of Technology, Xi'an 710048, China; 2190220081@stu.xaut.edu.cn (P.G.); 2200221159@stu.xaut.edu.cn (Y.G.); jingjingliu@xaut.edu.cn (J.L.); dengxinhua@xaut.edu.cn (D.H.)

**\*** Correspondence: wangjun790102@xaut.edu.cn

**Abstract:** Cloud droplets size distribution (DSD) is one of the significant characteristics for liquid clouds. It plays an important role for the aerosol–droplet–cloud mechanism and variation in cloud microphysics. However, the minuscule sampling space is insufficient for the observation of whole DSD when using high-magnification optical systems. In this paper, we propose an observation method for cloud droplets ranging from 2 to 16 μm, by which the balance relationship between sampling space and optical magnification is realized. The method combines an in-line digital holographic interferometer (DHI) with the optical magnification of 5.89× and spatial stitching technique. The minimum size in DSD is extended to 2 μm, which improves the integrity of size distribution. Simultaneously, the stability of DSD is enhanced by increasing the tenfold sampling volume of cloud droplets. The comparative experiment between the in-line DHI and fog monitor demonstrates that the DSD obtained by this method is reliable, which can be used for the analysis of microphysical parameters. In the Beijing Aerosol and Cloud Interaction Chamber (BACIC), the observation results show that the size of cloud droplets follows the Gamma distribution, which is consistent with the theoretical DSD. The results of cloud microphysical parameters indicate that each pair of parameters has a positive correlation, and then the validity of observation method is confirmed. Additionally, the high-concentration aerosol condition significantly mitigates the effect of random turbulence and enhances the robustness of the microphysical parameter data.

**Keywords:** liquid cloud; size distribution; microphysical characteristics; cloud droplet observation; digital holography

## 1. Introduction

Cloud microphysics, which is used for the investigation of precipitation mechanisms and construction of climate models [1–5], is one of the important parameters for investigating clouds and climates. In order to simulate the cloud processes under different conditions, many research institutions have established different types of cloud chambers, such as the Beijing Aerosol and Cloud Interaction Chamber (BACIC), Michigan Technological University Π chamber, and European Council for Nuclear Research cloud chamber [6,7]. The reproducible experimental processes are carried out in cloud chamber with variable experimental conditions of temperature, humidity, and number concentration of aerosols [8]. Desai et al. [9] observed the cloud droplets size distribution (DSD) and number concentration in a Π cloud chamber with an in-line digital holographic interferometer (DHI), in order to observe the DSD variation caused by condensation growth processes. However, due to the limit of optical magnification (2.85×), the observed smallest droplet size was 7 μm. Chandrakar et al. [10] used a Phase-Doppler Interferometer to measure the diameter of cloud droplets and estimated the DSD in a Π cloud chamber. The author suggested that the number concentration of smaller droplets was underestimated. During the measurement, the smallest credible droplet size was 7.5 μm. In the natural environment, Beals et al. [11]

measured centimeter-scale heterogeneous cloud mixing based on digital holography. The DSD was divided into three channels, and the smallest credible droplet size was 10 µm. However, the size of liquid cloud droplets is usually 2–100 µm [12–15]. Here, the size is the diameter of a droplet. Especially with the increase in the number of cloud condensation nuclei (CCN) [16–25], the number of droplets close to 2 µm is increased. Hence, in order to investigate physical processes such as condensation and collision in the cloud, 2 µm droplets should be measured in a cloud chamber or high-altitude cloud layer to acquire the complete DSD.

A Charge Coupled Device (CCD) or Complementary Metal-Oxide-Semiconductor Transistor Device (CMOS) is used to record digital holograms [26–30]. The recording processes and processing procedures of holograms are greatly simplified by using a DHI [31–36]. Due to the advantages of fast, real-time, non-contact, and full-field measurement, DHI is regarded as a technology for simultaneous observation of dynamic multi-parameter physical fields. It has been successfully applied in the fields of particle three-dimensional motion [37–42], biological cell imaging [43–47], holographic subsurface radar [48,49], and other fields. In this paper, the in-line DHI with high optical resolution is proposed to observe the droplets ranging from 2 to 16 µm in the BACIC. In order to increase the proportion of small droplets, the high concentration CCN is added to BACIC. In the absence of the droplets ranging from 2 to 6 µm, the experimental results show that the pattern of microphysical parameters is changed, and the reliability of observation data is reduced. Some important patterns are ignored, and even the opposite patterns are caused in cloud microphysical processes. The research contributes to data support for the theoretical analysis of cloud physical processes and development of parameterization schemes. Cloud microphysical data is provided for research in the fields of weather, climate, artificial weather modification, and atmospheric chemistry.

## 2. Theory

In in-line DHI, the mutual interference between a droplet and its twin image is small. When a plane wave is used to irradiate droplets, the diffracted light of the droplets (as the object light) interferes with the undisturbed plane wave (as the reference light) and is recorded as a digital hologram by a CMOS. By using numerical reconstruction, a two-dimensional amplitude distribution with the reconstruction distance $z_r$ can be obtained on the reconstruction plane, given by [50]

$$U_R(u,v) = \frac{1}{j\lambda} \iint\limits_{\infty} R(x,y) I_H(x,y) \frac{\exp\left(jk\sqrt{(u-x)^2 + (v-y)^2 + z_r^2}\right)}{\sqrt{(u-x)^2 + (v-y)^2 + z_r^2}} dx dy, \quad (1)$$

where, $\lambda$ is the wavelength, $R(x,y)$ is the reference light, $I_H(x,y)$ is the intensity of interference fringes on the recording medium, $k$ is the wave number. When $z_r = z_i$ ($i = 1, 2, 3, \dots$ ), the droplets with the reconstruction distance $z_i$ in the reconstructed hologram are focused, and other droplets are out of focus.

However, it is more suitable for batch processing to sample the 12 mm length on the $z$-axis at equal interval of 0.35 mm. So, $z_r$ can be regarded as a constant set in advance. The reference light of the in-line DHI is a parallel beam, so $R(x,y) = 1$. With the convolution operation, the integral of Equation (1) can be expressed as [50,51]

$$\begin{cases} U_R(u,v) = I_H(x,y) \otimes g(x,y) \\ g(x,y) = \frac{1}{j\lambda} \frac{\exp\left(jk\sqrt{x^2+y^2+z_r^2}\right)}{\sqrt{x^2+y^2+z_r^2}} \end{cases} . \quad (2)$$

Since the convolution operation in Equation (2) is still complicated, the frequency domain multiplication operation can be used instead to reduce the complexity. Therefore, Equation (2) is expressed by Fourier transform as

$$U_R(u,v) = \mathcal{F}^{-1}\{\mathcal{F}[I_H(x,y)]\mathcal{F}[g(x,y)]\}. \tag{3}$$

The 35 reconstruction images $U_R(u,v)$ obtained by sampling are used to identify the focused droplets to obtain the three-dimensional positions and diameters of droplets. The number of droplets in the sampling space is counted as $N_c$, and the number concentration $N$ can be expressed as $N_c$ divided by volume $V$. Diameters of all droplets are averaged, the mean volume diameter (MVD) can be obtained as

$$\text{MVD} = \sum_{i=1}^{N_c} l_i, \tag{4}$$

where, $l_i$ is the diameter of the $i$-th droplet and $N_c$ is the number of droplets in sampling space. Through the statistics of droplet size information in 35 reconstructed images, the actual DSD can be acquired. The above distribution is considered to obey the Gamma distribution. It can be expressed by a corrected Gamma distribution function $n(l)$, given by

$$n(l) = al^5 \exp(-bl^c), \tag{5}$$

where, $a$, $b$, and $c$ are three fitting parameters of $n(l)$. They are assigned by the principle of least square method, and the residual sum of squares of Equation (5) is the smallest. The effective diameter (ED) can be calculated based on $n(l)$, given by

$$\text{ED} = \frac{\int_0^\infty l^3 n(l)dl}{\int_0^\infty l^2 n(l)dl}, \tag{6}$$

Here, MVD and number concentration $N$ are directly obtained from the holographic measurement result, which does not depend on the fitting result of $n(l)$. Hence, MVD is used for the analysis of microphysical variations instead of ED, and $n(l)$ is only used to verify the integrity of size distribution. The diameter of each droplet in sampling space can be used to calculate the liquid water content (LWC) of the current environment [52], given by

$$\text{LWC} = \sum_{i=1}^{N_c} \frac{4\pi\rho_c}{3V} \left(\frac{l_i}{2}\right)^3, \tag{7}$$

where, $\rho_c$ is the density of water and $V$ represents the volume of sampling space.

## 3. Experimental Setup

The evolution processes of cloud can be simulated by expanded form in the BACIC. By continuously reducing pressure and temperature, the aerosols acting as CCN become cloud droplets under high relative humidity conditions. In the experiment, a high-resolution DHI is used to measure the microphysical parameters in the BACIC, as shown in Figure 1. The Condensation Particles Counter (CPC) is used to measure the aerosol number concentration and the DHI is placed on the experimental platform at the bottom. The cloud chamber is a cylindrical structure (the diameter of 2.6 m and volume of 70.5 m³) with approximately uniform distribution of cloud droplets at the bottom. In DHI, a frequency-doubled solid-state laser produced by the German LASOS company is used as the light source. The wavelength is 532 nm and the output power is 120 mW. In order to ensure the stability of the laser in complex environments, a beam splitting prism with a transmission–reflection ratio of 99:1 is used to divide the beam into two parts. In total, 1% of the beam is reflected to the photodiode (PD) to measure the energy fluctuation of the laser caused by environment temperature and feed it back to the laser control system. The promotion integration differentiation (PID)

algorithm dynamically adjusts the drive current of the laser. A total of 99% of the light beam is collimated and expanded acting as a broad beam using three aspherical lenses (L1–L3), and irradiates the measured cloud droplets. The diameters of L1, L2, and L3 are 10, 30, and 40 mm, respectively. The interference image formed by the droplet diffracted light and the unmodulated transmitted light is zoomed 5.89 times by the microscope lens (ML). The working distance is 20 mm from the ML. The enlarged image is recorded by a CMOS. The minimum exposure time and resolution are 10 µs and 5120 H × 5120 W, respectively. The minimum sampling frequency of CMOS at full resolution is 15 Hz. The chip of CMOS is produced by China Daheng (Group) Company. In order to reduce the influence of field curvature and distortion, the pixels of 3200 W × 3200 H in the central area of CMOS are used. Therefore, the sampling area in the *x-y* section is 1.35 × 1.35 mm, the sampling distance along the beam direction (*z*-axis) is 12 mm, and the zero point of the *z*-axis is set on the focal plane of ML. A polarizer (P) is placed behind the ML to improve the contrast of interference fringes.

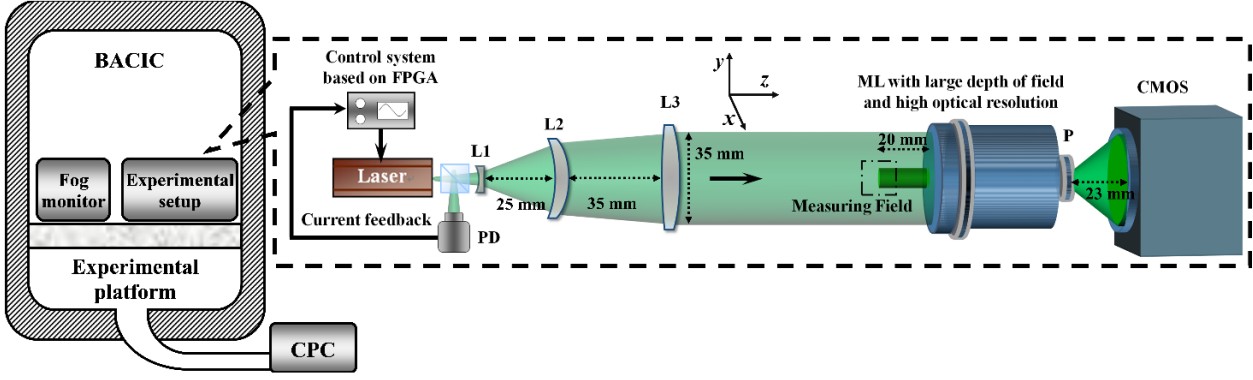

**Figure 1.** The measurement method of the DHI.

Due to the limitation of target surface size of commercial cameras, high resolution and sample volume are a contradictory relationship. By limiting the sampling volume, the holographic device achieves a high resolution. To ensure complete the observation of 2 µm particles, the sampling volume is reduced to 0.2187 cm$^3$. The USAF1951 resolution board is observed to verify the optical resolution of the DHI. Figure 2 shows the reconstructed images at −6, 0, and 6 mm on the *z*-axis. The smallest distinguishable line pairs in Figure 2a–c are labeled 7–6, and the corresponding line width is 2.19 µm. By observing standard particle glass plates with diameters of 2, 10, and 20 µm, the accuracy of diameter identification is verified. The particles of 2 and 10 µm are arranged in a square array at an interval of 8 times the diameter, and the interval of 20 µm particles is set to 12 times the diameter. The image algorithms of corrosion and expansion are used for particle identification, which results in a deviation of 1 equivalent pixel. So, the theoretical error of diameter measurement is ± 0.42 µm. The reconstructed image of the particle glass plate at 6 mm from the focal plane is shown in Figure 3. In Figure 3a–c, 795, 236, and 36 particles are identified, and the measurement uncertainties are (2.05 ± 0.53) µm, (10.12 ± 0.86) µm, and (19.89 ± 1.24) µm, respectively. In the cloud chamber, the number of particles in the sampling space of DHI is usually less than 50, so the accuracy of cloud droplet identification is higher than the particle glass plate.

In order to promote the application of in-line DHI in meteorological detection and atmospheric cloud microphysics research, it is necessary to carry out various comparison experiments with recognized cloud droplet detection instruments. The fog monitor [53–60] is widely used in mountain stations and cloud chambers to observe the cloud and fog processes. The measurement results of fog monitors are considered reliable by the majority of researchers. Therefore, we chose the fog monitor as the reference device to verify the accuracy of the holographic device in the measurement of cloud droplet size, number concentration *N*, and other parameters. In this cloud chamber measurement experiment,

the holographic device and the fog monitor are placed 15 cm apart on the experimental platform at the bottom of the cloud chamber. Due to the fact that the distance between two devices is narrow, it can be considered that the cloud droplet groups measured by two devices are highly similar. Then, the most important item in the comparison of measurement results is the size distribution of cloud droplets. The DSDs of fog monitors and in-line DHI are divided by the sampling volume for normalization. Additionally, the DSDs of two devices are nearly identical, as illustrated in Figure 4. The measurement result of the fog monitor is represented by the bold black line and the measurement result of the in-line DHI used in the experiment is represented by the green bar graph.

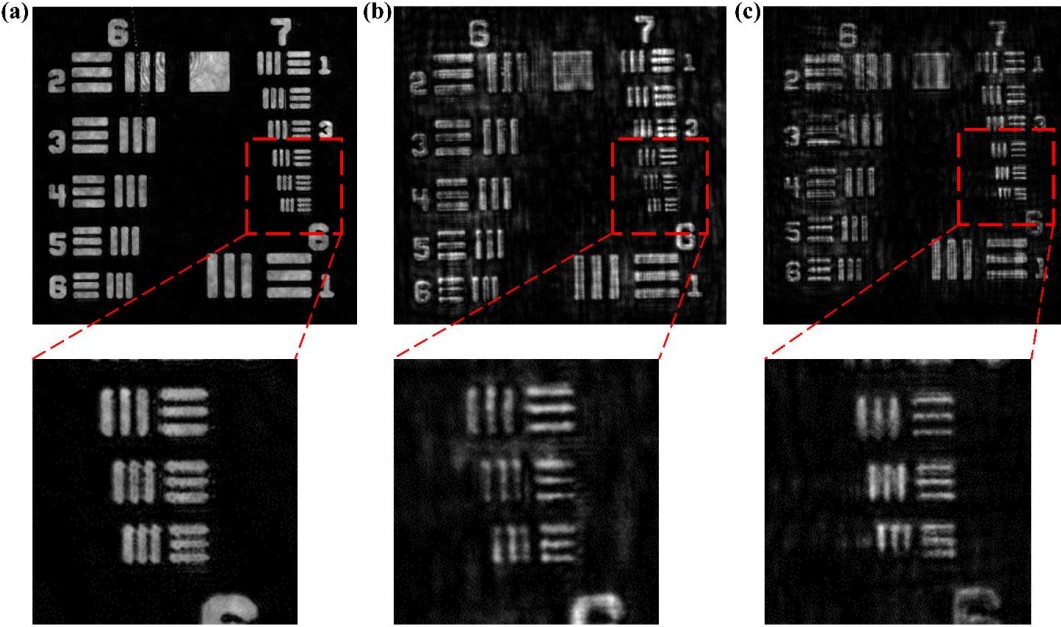

**Figure 2.** The reconstructed image of the USAF 1951 resolution plate. (**a**) At 0 mm. (**b**) At 6 mm. (**c**) At −6 mm.

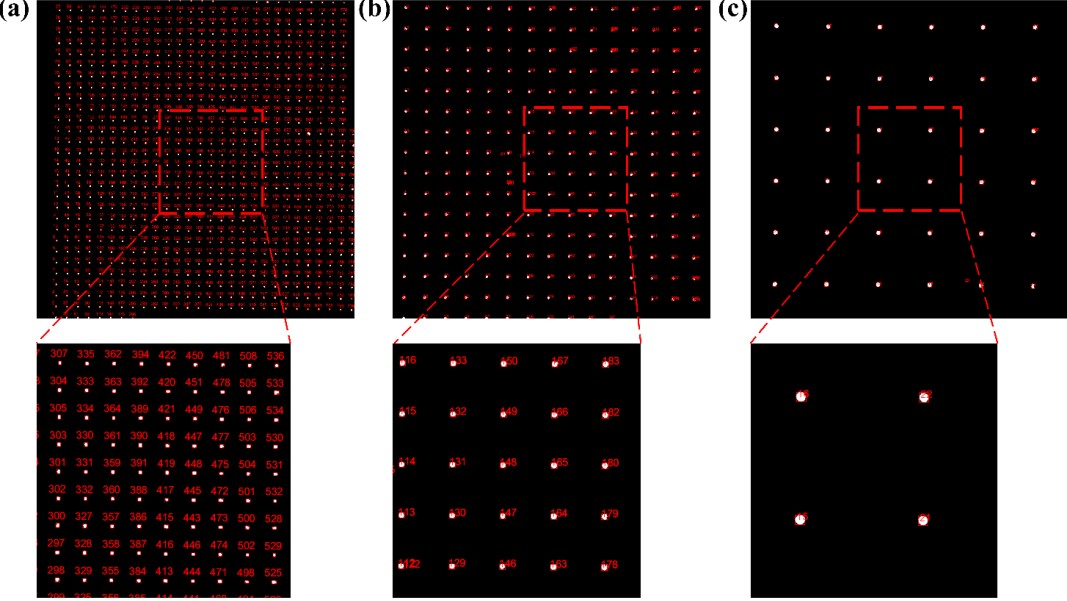

**Figure 3.** Experimental results of the standard particle glass plate at the distance of 6 mm from the focal plane. (**a**) 2 μm particles. (**b**) 10 μm particles. (**c**) 20 μm particles.

Typically, researchers are interested in the size range corresponding to the maximum value of the DSD, which is used to denote the center of the size distribution. The centers of the size distributions in Figure 4a–d are all 6–7 μm, indicating that the four size distributions collected exhibit similar characteristics. On the one hand, the DSD of two instruments is concentrated at 4–8 μm, as a result of cloud droplet competition for growth. Therefore, the observed cloud droplet exhibits a normal distribution or Gamma distribution. Here, the competitive growth refers to the fact that, as the cloud chamber expands, the majority of CCNs absorb water vapor at the same rate in order to grow in volume. Cloud chamber expansion is achieved by extracting the internal gas. Due to the gas exchange in the extraction processes, a large amount of turbulent flow field is introduced into the cloud chamber. Simultaneously, the entrainment of dry and wet air is also formed. A part of CCN is accelerated by the turbulent flow field, while another part was entrained by moist and dry air, thereby slowing the growth rate. On the other hand, two devices have similar characteristics in the gradient change in the size distribution. The ascending gradient of the cloud droplet size distribution remains slow and uniform below 6 μm, while the descending gradient increases above 6 μm. However, there are slight differences in the results of the two devices due to the difference in measurement principle and observation volume. Compared with the DSD shape of the fog monitor, the DSD shape measured by the holographic device has a larger gradient of rising and falling. Simultaneously, the measurement value of the holographic device in the range of 3–4 μm is less than that of the fog monitor. Except for Figure 4c, the measured values of the holographic device in the range of 8–12 μm are less than that of the fog monitor. These differences can be explained by the measurement principle of the fog monitor. This modifies the spatial distribution of cloud droplets to a certain extent, bringing the resultant DSD closer to statistical distribution, and improving the uniformity of the change in the entire DSD shape. The measurement method of DHI does not alter the spatial distribution of cloud droplets. The gradient of size distribution varies significantly due to the volume ($0.219 \text{ cm}^3$) constraint of the sampling space. As a result, each device has distinct advantages and disadvantages. In summary, the DSDs of the DHI and fog monitor are nearly identical, which satisfies the requirements for cloud droplet observation and subsequent cloud microphysical parameter analysis.

The size distribution of cloud droplets can be used to determine the state of cloud droplets at various times. However, the analysis of these distribution characteristics in time series should be performed using the parameter of MVD. In meteorological research, more attention is paid to the formation and dissipation processes of cloud droplets. The dissipation stage lasted approximately 60 s, and 600 holograms are processed to obtain the time-series contrast curve of MVD, as shown in Figure 5. Here, the sampling frequency of the holographic device is 10 Hz, while the sampling frequency of the fog monitor is 1 Hz. To facilitate data comparison, the frequency of the holographic device is reduced to 1 Hz by averaging 10 data points, as shown in Figure 5a. The MVD curves of both devices reflect the dissipating pattern of cloud droplets. Within 60 s, the MVD of both devices followed a similar downward pattern, demonstrating the reliability of holographic device. The MVD of the droplet spectrometer is generally 2 μm higher than that of the holographic device during the last 20 s. This is because the holographic device has a "0 value" during the 20 s, as indicated by the three points with ordinates of 0 in Figure 5a. The binary transitions between 0 and a fixed value are referred to as "zero-sampling". To more clearly represent this "zero-sampling", the sampling frequency of DHI is reduced from 10 to only 5 Hz, as shown in Figure 5b. From the 30th to the 60th second, there are 48 "zero-sampling" points out of 150 data points, which is quite a novel phenomenon. Combined with the related research of Desai et al. [9] in turbulent cloud entrainment, they proposed that the change in the average droplet diameter at the entrainment boundary is binary, abruptly jumping between 0 and a fixed value, which corresponds to the "zero-sampling" effect in Figure 5b. As a result, the dissipation of cloud droplets leads to the appearance of a large number of entrained and mixed regions in the BACIC. Here, the phenomenon of "zero-sampling" indicates that the distribution of cloud droplets is extremely uneven during

the cloud dissipation. It can be considered that the spatial distribution characteristics of cloud droplets are still retained in the cloud microphysical parameters obtained by three-dimensional measurement. Therefore, the DHI device also contributes to the study of small-scale phenomena such as clusters, voids, and random condensation in clouds, which is an exciting area of research to pursue.

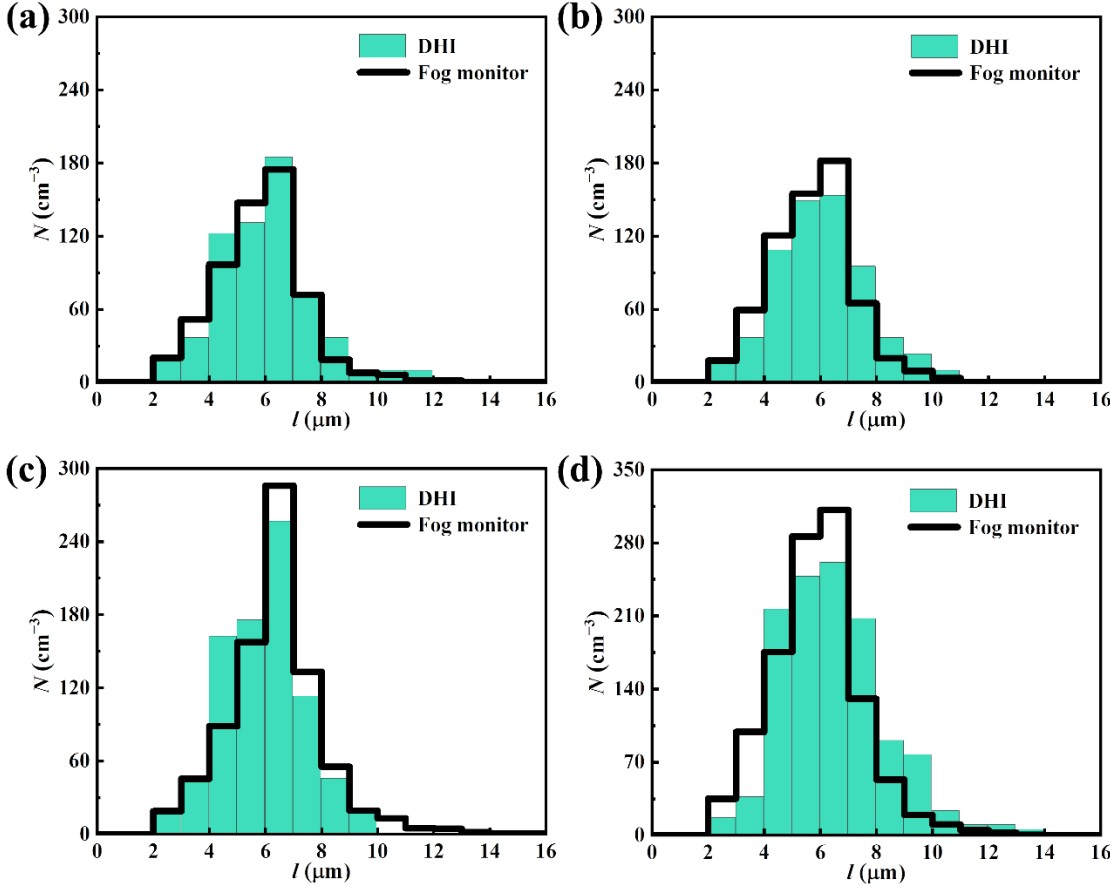

**Figure 4.** Comparison of DHI and fog monitor on the DSD. (**a**) Size distribution at 1.0 min. (**b**) Size distribution at 4.0 min. (**c**) Size distribution at 2.0 min. (**d**) Size distribution at 3.0 min.

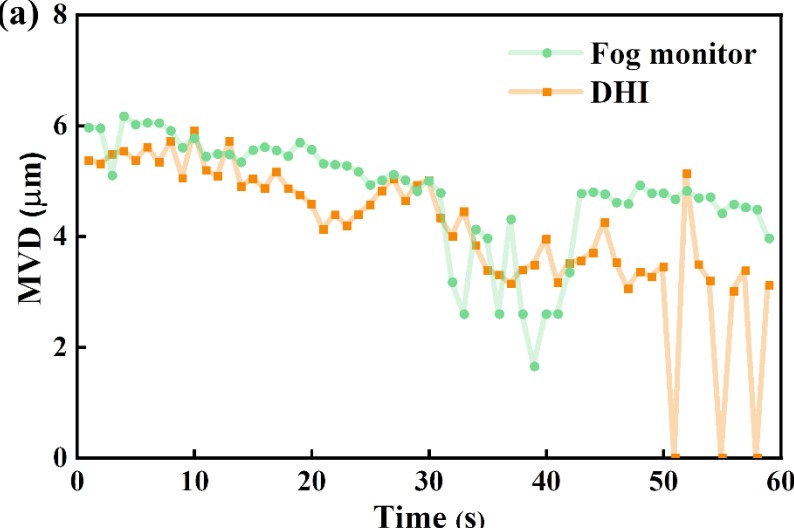

**Figure 5.** *Cont.*

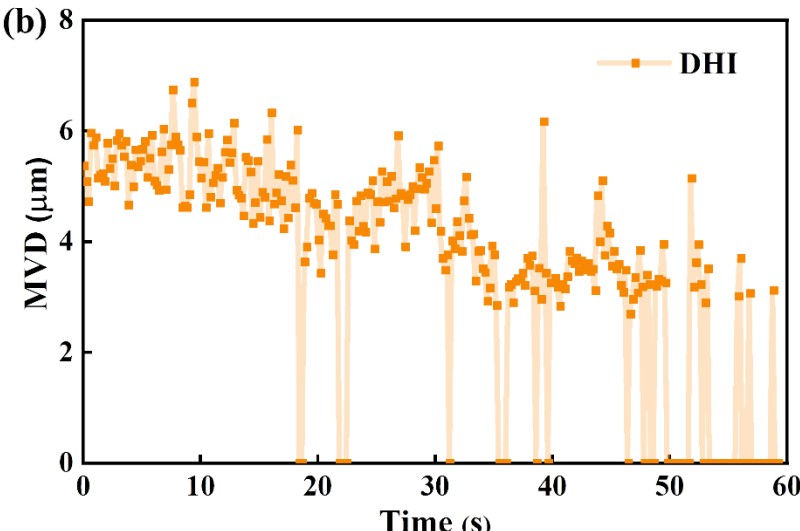

**Figure 5.** Comparison of DHI and fog monitor in relation to MVD during the last 60 s of the cloud processes. (**a**) The sampling frequency of both DHI and fog monitor is 1 Hz. (**b**) The MVD results of DHI with sampling frequency of 5 Hz.

## 4. Results Analysis and Discussion

### 4.1. Size Distribution

The cloud chamber is used as a device for simulating the cloud processes, and the external gas is inhaled by a filter device to fill the interior with aerosol particles. Experiments in the cloud chamber are carried out to verify the accuracy of cloud droplet measurements and the reliability of the software for real-time reproduction of digital holograms. By spraying small liquid droplets (mean diameter 50 µm) above the cloud chamber, the water vapor content in the BACIC is increased. After the spraying operation is completed, the pressure is reduced at a rate of 61 hPa per minute, and cloud droplets are generated 20 s later. The cloud processes last about 5 min, the pressure is reduced by 307 hPa, and the temperature is reduced by 9.5 K. The experimental conditions of the cloud chamber in three experiments are set to the same value, and the environmental parameters of the initial state are relative humidity (95%), temperature (296.2 K), and pressure (998 hPa). The sampling frequency of the in-line DHIis set to 10 Hz for stable image acquisition. In the experiment, the first 10 holograms are averaged to obtain the background hologram of the 11th hologram. By subtracting the obtained background image from the 11th hologram, the noise such as background light is removed. Except for the first 10 holograms, every hologram is processed by the above method. The processes from the hologram to reconstructed image and droplet recognition are shown in Figure 6. During the sampling processes of 12 mm in the *z*-axis direction at a 0.35 mm interval, the droplet recognition algorithm is used for cloud droplet detection in 35 reconstructed images. Here, the recognition processing of five cloud droplets in the hologram is shown in detail. By droplet recognition, the *x*, *y*, and *z* axis coordinates and diameters of droplets are obtained. Software processes of real-time reconstruction are shown in Figure 7. The steps of droplet recognition are marked by dashed lines, which correspond to the schematic diagram of Figure 6.

DHI is used as a three-dimensional display technology for cloud droplet measurements. The three-dimensional position and size of droplets are obtained at the same time, as shown in Figure 8a–k. DSD can be obtained by counting the diameters of all droplets, and the DSD shape reflects the characteristics of droplet size. The 5 min cloud processes in the first experiment are recorded, and the DSD at the 90th second is provided in Figure 8m. The fitting curve follows the Gamma distribution, and the area between 2 and 6 µm exceeds half of the total area. Hence, for environments with high aerosol number concentrations, the full distribution of DSD can be improved by using an in-line DHI to observe droplets greater than or equal to 2 µm.

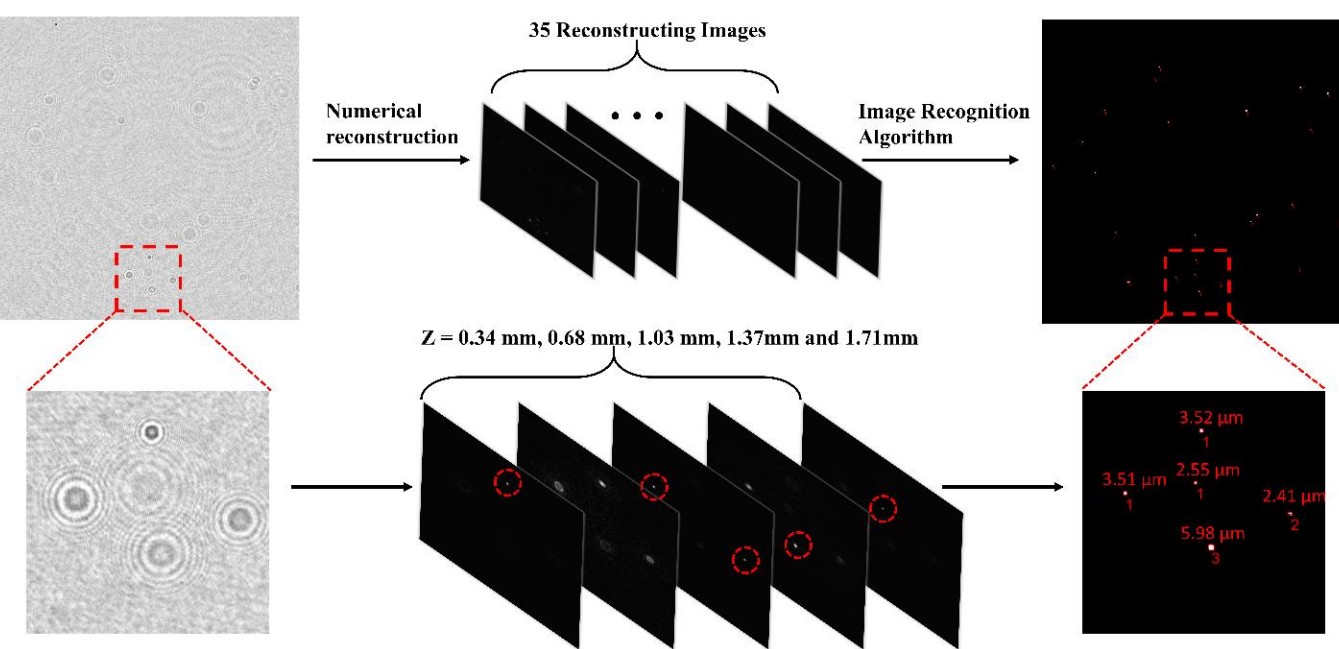

**Figure 6.** The schematic diagram of droplet recognition.

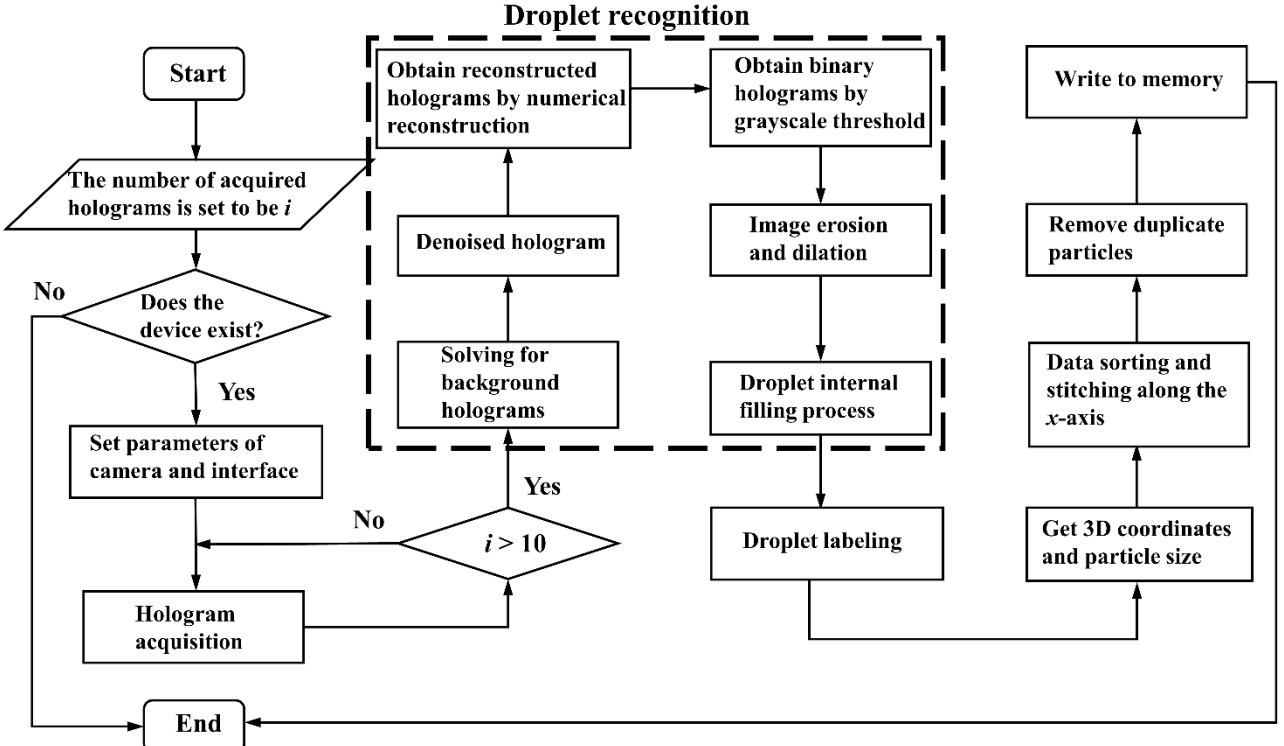

**Figure 7.** Software processes of real-time reconstruction.

According to the statistical results of 3000 holograms at the first experiment, the number of droplets in each hologram is found to be between 30 and 50. There are not enough overall size characteristics to be represented. The 10 sampling areas are spliced along the positive *x*-axis in time series to expand the sampling space and reduce the randomness of data, as shown in the sampling area of Figure 8k. The spliced area is a rectangular parallelepiped of $13.5 \times 1.35 \times 12$ mm, and the sampling frequency is reduced to 1 Hz. The DSD is usually divided into 1 μm channels according to cloud microphysics

theory, as shown in Figures 8m and 9. The experimental processes were observed in three experiments for analyzing the influence of aerosol number concentration at the DSD.

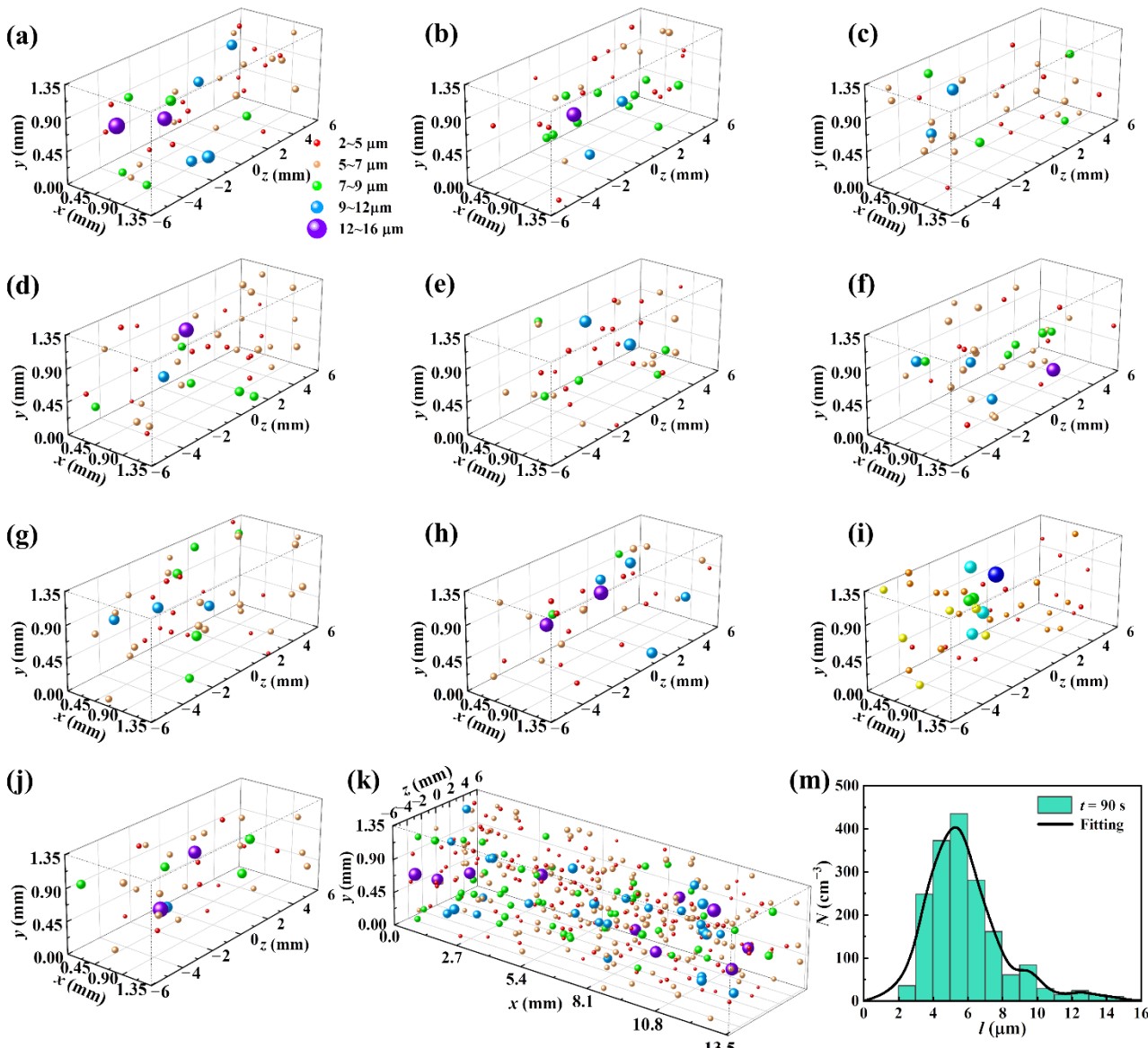

**Figure 8.** The three-dimensional distribution and DSD in the 90th second. (**a**–**j**) show the 10 three-dimensional distributions of cloud droplets in the detection area within 1 s. (**k**) The three-dimensional distribution fused by 10 distributions. (**m**) The DSD corresponding to the 90th second.

In three experiments, aerosol number concentrations $N_a$ measured by CPC are 10,390, 9150, and 7370 cm$^{-3}$, respectively. The CCN in the cloud chamber is reduced from the first experiment to the third experiment. By distinction of size, the DSD is divided into smaller droplets (2 µm–$l_{mid}$), medium droplets ($l_{mid}$–$l_{mid+1}$), and oversized droplets ($l_{mid+1}$–16 µm), as shown in Figure 9a. Figure 9a–c show the three size distributions at equal intervals in the first experiment. The value of $l_{mid}$ is 5 µm, and the proportion of smaller droplets is significantly higher than oversized droplets. This is due to the reduced average water vapor absorption of the droplets at high aerosol number concentration. The $l_{mid}$ increased to 6 µm in the second experiment as shown in Figure 9d–f. Under the condition that $N_a$ is reduced by 11.9%, the proportion of smaller droplets is close to oversized droplets. In the third experiment, the aerosol number concentration of the experiment is reduced by 19.4%, and $l_{mid}$ remains at 6 µm. However, the proportion of smaller droplets is already

less than that of oversized droplets, as shown in Figure 9g–i. The above results describe that the right shift of DSD is caused by the decrease in $N_a$, and the proportion of smaller droplets is gradually reduced. This indicates that the competitive growth of droplets is weakened as the total number of droplets decreases, resulting in the increase in droplet size. Therefore, compared with $l_{\mathrm{mid}}$, the change in $N_a$ can be more reflected by the proportion of smaller droplets.

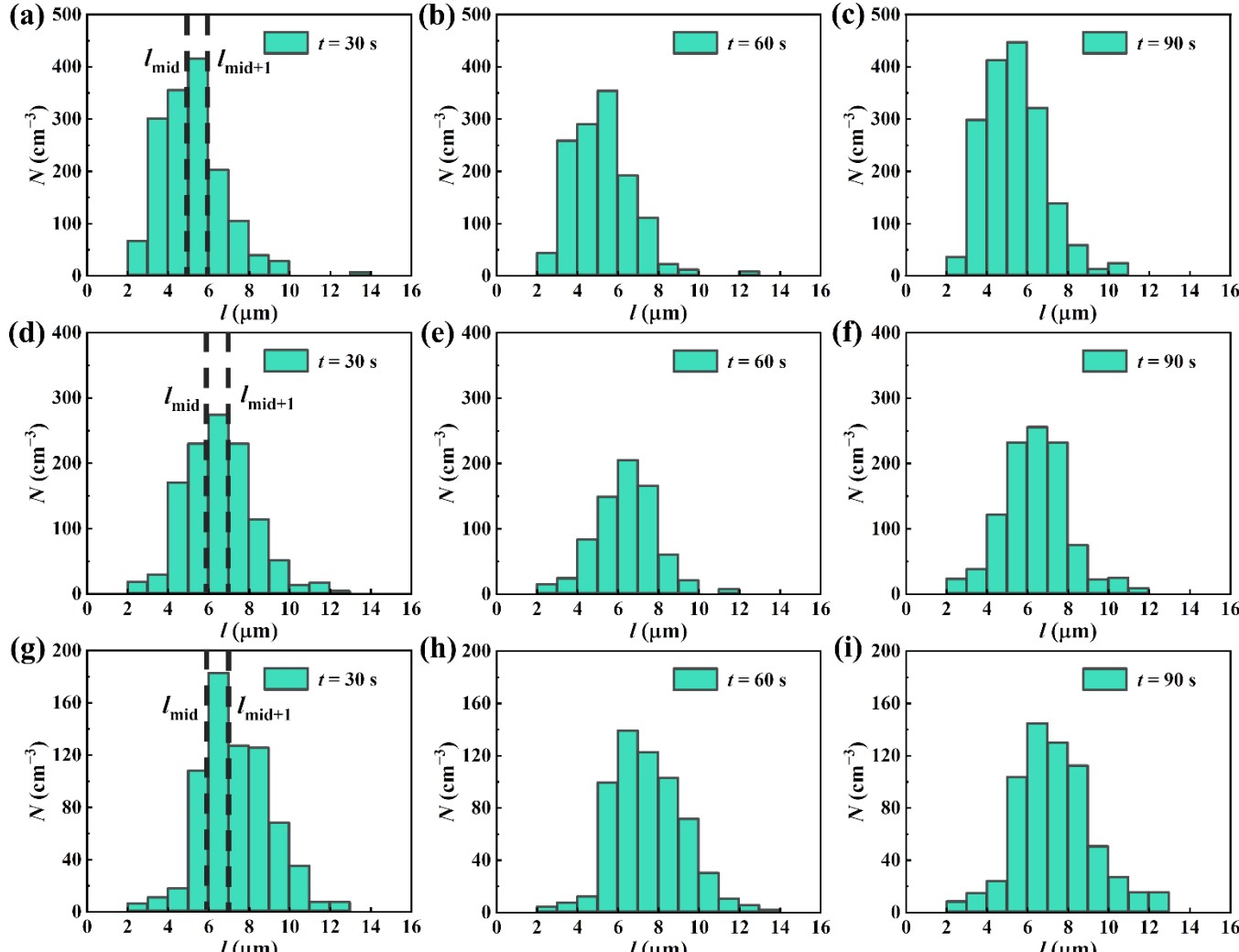

**Figure 9.** The DSD in cloud chamber. (**a–c**) show the size distribution with $N_a = 10{,}390\ \mathrm{cm}^{-3}$. (**d–f**) show the size distribution with $N_a = 9150\ \mathrm{cm}^{-3}$. (**g–i**) show the size distribution with $N_a = 7370\ \mathrm{cm}^{-3}$.

The contribution of each size channel is counted. From Figure 10a, the number concentration is represented by the height of histogram, and the range of mean diameter is reflected in the line graph. The overall number concentration is reduced, and the mean diameter is slowly increasing. As for 2 μm–$l_{\mathrm{mid}}$, Figure 10b shows the proportion of the divided three types of droplet sizes. The proportion of smaller droplets in the entire DSD is reduced from 45.47% to 38.70% in the second experiment, and then reduced to 21.26% in the third experiment. Therefore, the statistical result shows that the proportion of smaller droplets is positively correlated with $N_a$.

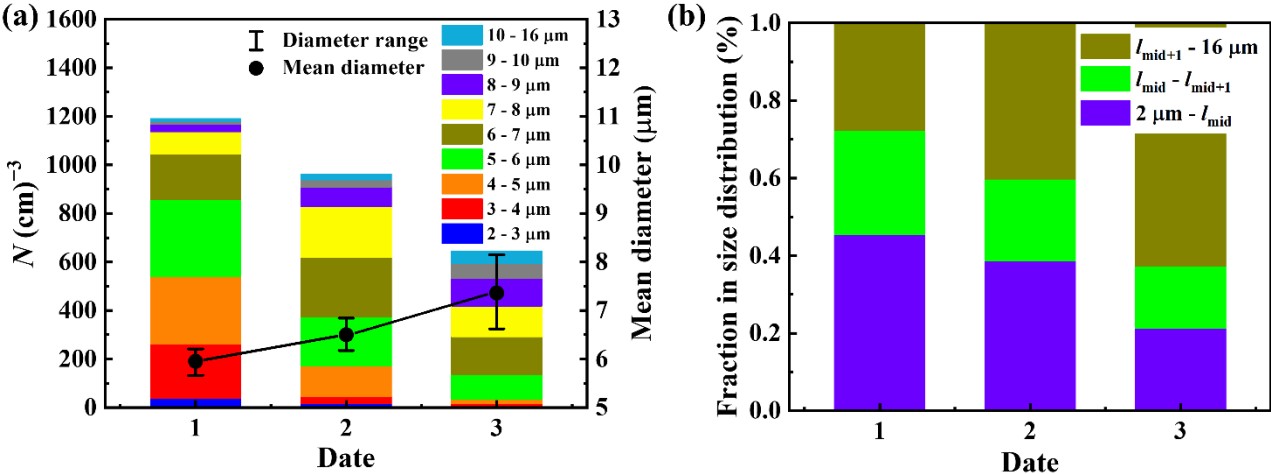

**Figure 10.** The average statistical results of DSD in three experiments. (**a**) The number concentration distribution of nine size channels. (**b**) The proportion of three types of droplets: smaller, medium, and oversized droplets.

### 4.2. Microphysical Parameters

The droplet size characteristics at the current moment can be reflected by the DHI in the real-time detection processes. However, the expanding cloud processes last 5 min. The entire experimental processes obviously cannot be represented by the DSD within 1 s. Therefore, the time-series variations in the DSD are transformed into microphysical variations, such as the number concentration $N$ and the LWC. In Figure 11a,c, the data is low-pass filtered to show the pattern of the whole processes. As the $N$ of cloud droplets is increased and decreased by fluctuations, the processes of cloud droplet growth and dissipation are reflected. Due to the weak upward and downward patterns, the original data of MVD is used, as shown in Figure 11b. The comparison results of parameters at the three concentrations show that the decrease in $N$, the increase in MVD, and the decrease in LWC are caused by the decrease in $N_a$. At the same time, the pattern of LWC is basically the same as $N$, and the rate of increase or decrease is related to MVD. However, due to the long-term growth processes (2.1–2.8 min) at $N_a = 7370$ cm$^{-3}$, the values of other LWC curves are exceeded, as shown in Figure 11c. It is worth noting that the MVD curve rises rapidly at 2.2–2.8 min, as shown in Figure 11b. This indicates that the monotonic growth of LWC is caused by the volatility growth of MVD. Here, the ratio of standard deviation to average value is usually used to characterize the deviation degree of data, denoted as $S$. As the proportion of 2–6 μm droplets is reduced, the $S$ of $N$ is 0.198, 0.248, and 0.335, the $S$ of MVD is 0.058, 0.066, and 0.132, and the $S$ of LWC is 0.162, 0.259, and 0.416, respectively. Hence, the decrease in the measurement stability of cloud microphysical parameters is caused by the decrease in the proportion of 2–6 μm droplets.

For the whole cloud processes, the growth and dissipation processes of cloud droplets are usually divided by the rising and falling patterns of $N$. Due to the influence of the turbulent field and the limitation of the sampling volume, multiple wave crests appear on the time series curve of N. Here, the maximum value in the wave crest is considered as the sign of the end of the growth phase, and its corresponding time is the duration of cloud droplet growth. With the increase in $N_a$, the growth time of cloud droplets under the three aerosol concentrations is 2.9, 2.4, and 1.5 min, respectively. This shows that, with the same initial water vapor content, the greater the number of CCNs per unit volume, the easier it is to reach the saturation state of water vapor absorption. Here, the MVD curve maintained a downward pattern between 4.0 and 6.0 min, especially the green curve corresponding to the lowest concentration. Therefore, the last 60 s are used to compare holographic devices with fog monitors.

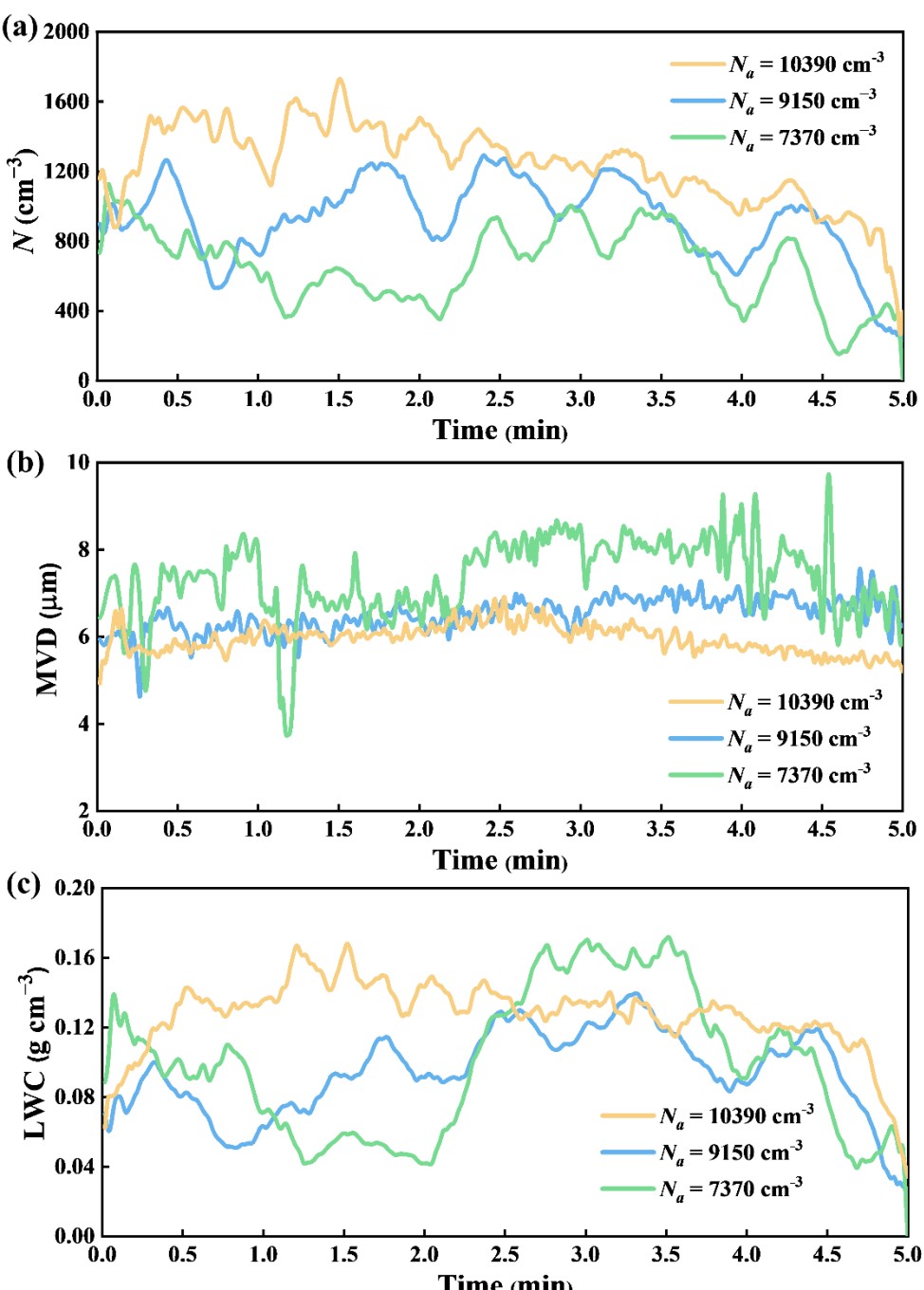

**Figure 11.** The microphysical parameters of cloud droplets under the different aerosol number concentrations. (**a**) Time-series variation in *N*. (**b**) Time-series variation in MVD. (**c**) Time-series variation in LWC.

In order to investigate the influence of the integrity of DSD on the microphysical processes, the 2–6 μm droplets in the third experiment are deleted, as shown in Figure 12. From the dotted area in Figure 12a, due to the random turbulence in cloud chamber, there are ascending, descending, and ascending processes at 1.0–1.2, 1.2–1.3, and 2.1–2.7 min, respectively. However, for the MVD with 2–6 μm droplets deleted, the steady variation at 2.1–2.7 min does not conform to the upward pattern of original curve. The absence of upward pattern is caused by the incomplete DSD, and the actual change in cloud is ignored. For the observing system, the absence of such a process is essentially considered the defect of data reliability. From the dotted area in Figure 12b, for the LWC with missing

droplets of 2–6 μm, the monotonic growth processes are reduced from 2.0–2.7 to 2.1–2.4 min. Compared with Figure 12a, the upward pattern is completely reversed to the downward pattern at 2.5 min. The faint fluctuations of 2.7–3.6 min are turned into two prominent wave crests. This shows that the interference of turbulence is enlarged by the absence of droplets. The above results indicate that the absence of 2–6 μm in DSD changes the actual pattern of cloud processes in some periods and reduces the reliability of observation data to cloud variations.

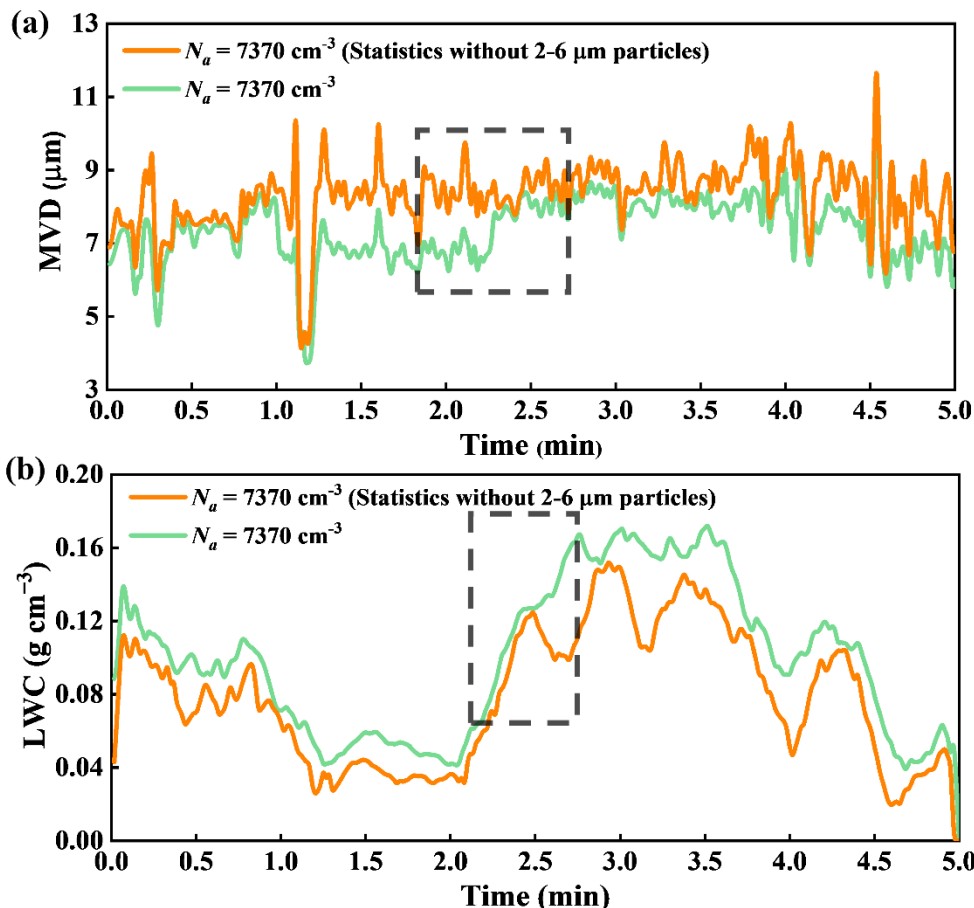

**Figure 12.** The changes in microphysical parameters at $N_a$ = 7370 cm$^{-3}$. (**a**) The original MVD and the MVD without 2–6 μm droplets. (**b**) The original LWC and the LWC without 2–6 μm droplets.

The microphysical parameters can reflect the quantity and size characteristics of particles. Among them, the $N$ and LWC are primarily used to describe quantitative characteristics, while the MVD, ED, and LWC are primarily used to describe size characteristics. The correlation between $N$ and LWC is shown in Figure 13a,b for $N_a$ of 9150 and 7370 cm$^{-3}$. According to the results of the linear fitting of LWC, there is a positive correlation between $N$ and LWC at both concentrations. In the analysis of Figure 11, the change pattern of $N$ and LWC curves is similar, which is a manifestation of positive correlation. It is worth noting that as $N$ increases, the fluctuation range of LWC is gradually increased. Especially under the condition that $N_a$ is 7370 cm$^{-3}$, the maximum fluctuation difference reaches 0.14 g cm$^{-3}$. Correlations of $N$ and LWC with $N_a$ of 9150 and 7370 cm$^{-3}$ are shown in Figure 13a,b. The results of the linear fitting reveal a positive correlation between ED and MVD. Unlike the fluctuation change in LWC, the fluctuation change in ED around the fitted curve is kept to a maximum of 1.3 μm in Figure 13c. In Figure 13d, this fluctuation is limited to a range of no more than 2.1 μm. In this case, the fluctuation of the curve represents a data bias, and the fitted curve represents an estimate of the data that approaches the true value. Thus, the

increase in aerosol concentration reduces the deviation degree of the data and improves the anti-interference ability of the data, which is also called the robustness.

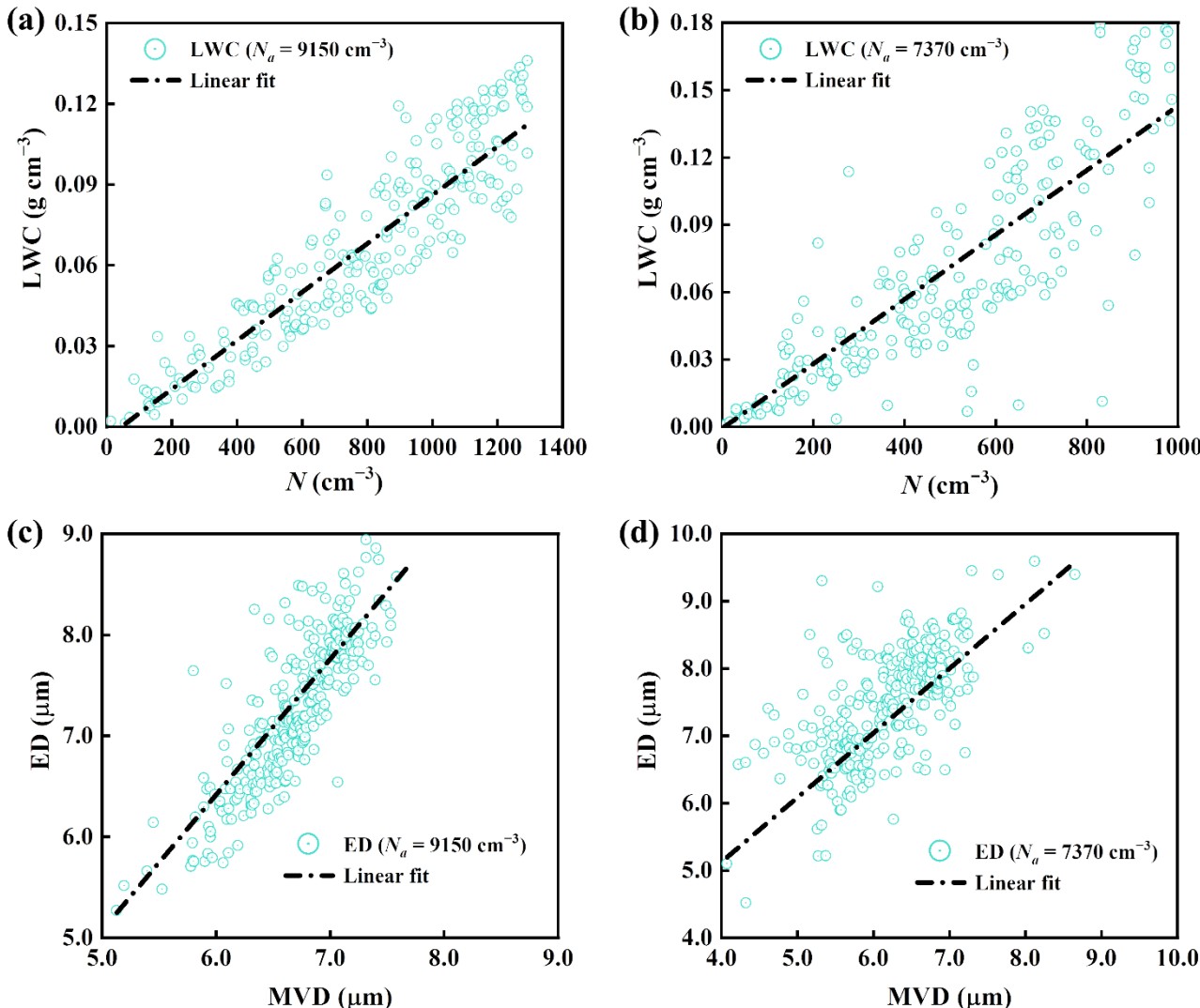

**Figure 13.** Microphysical parameter correlation curves are shown for $N_a$ of 9150 and 7370 cm$^{-3}$, respectively. (**a,b**) show the relationship between $N$ and LWC. (**c,d**) show the relationship between ED and MVD.

To conduct a qualitative analysis of the potential association between MVD and $N$, we classify the aerosol concentrations across three experiments into three categories based on their numerical magnitude: high, moderate, and low. In Figure 14, the fitting curve for MVD maintained an upward pattern as the number concentration increased, demonstrating a slight positive correlation between MVD and $N$. Notably, the data distributions at the three concentrations showed obvious stratification. There are two main reasons for the stratification. On the one hand, the values of $N_a$ differ by more than 1000 cm$^{-3}$ in three experiments. A large concentration gradient results in the stratification of MVD. On the other hand, the maximum deviation distance between discrete point and fit curve is less than 2 μm in Figure 14. The limited deviation distance also causes the stratification. In addition, there are some differences in the dispersion degree of data between three concentrations. When $N$ is in the range from 800 to 1100 cm$^{-3}$, the dispersion degree of data is increased as the aerosol concentration is decreased. With the high-concentration condition of aerosol, the fitted curve grows approximately linearly at $N$ in the range of 800–1400 cm$^{-3}$, which is consistent with the theoretically proportional relationship. Since

the fluctuation in the fitted curve is mainly affected by the uneven distribution under the action of turbulence, the high-concentration aerosol condition effectively weakens this effect and enhances the robustness of the data.

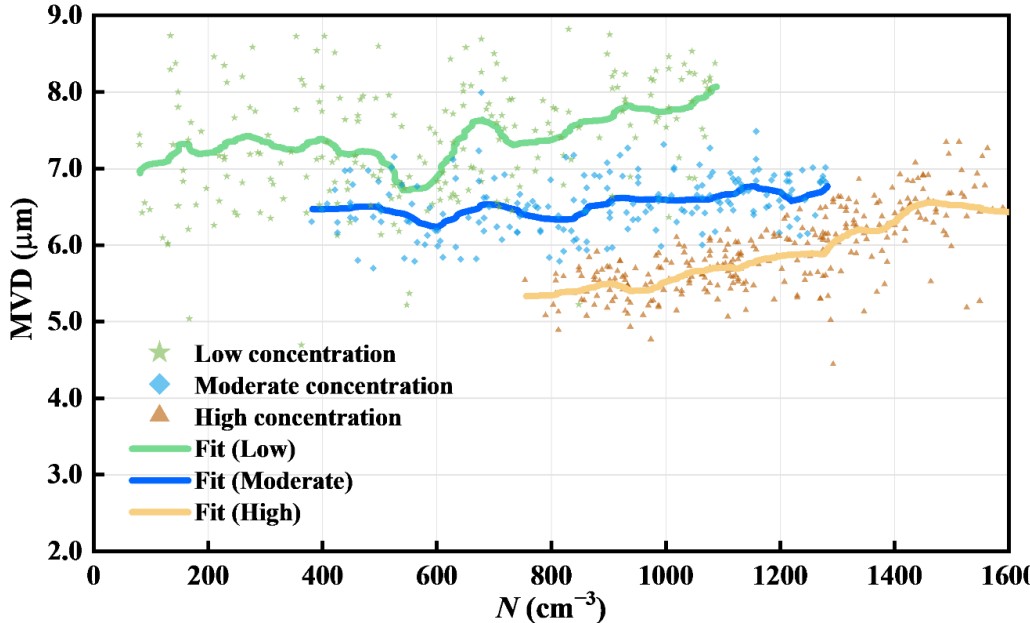

**Figure 14.** Correlation analysis between *N* and MVD at three different aerosol concentration levels: high, moderate, and low.

Here, the correlation analysis is performed on several cloud microphysical parameters in Figures 13 and 14. The concentration of aerosols directly influences the correlation between any two microphysical parameters. Regardless of whether Figures 13 or 14 is shown, the data distribution associated with the high concentration is denser, and the data deviation is smaller in the vicinity of the fitted curve. This indicates that DHI device may be better suited for particle detection at higher concentrations of particles. Naturally, there is a limit to the aerosol concentration inside the cloud chamber for cloud droplet observation experiments. Simultaneously, as DHI is a three-dimensional imaging device, the extremely high density of cloud droplets will result in increased particle overlap, lowering the accuracy of particle identification. Therefore, the initial aerosol concentration around 9150 cm$^{-3}$ is an optimal choice.

## 5. Conclusions

In the paper, the observation method based on DHI of the liquid cloud droplets with the size of 2–16 μm is proposed for extending the observed DSD in the BACIC. With the USAF 1951 resolution plate and the glass plate of 2 μm particles, the minimum reliable size observed by DHI is 2 μm. The sampling volume is increased tenfold by stitching three-dimensional space along the *x*-axis. After the particle identification, the 3D position and DSD of cloud droplets are obtained. The comparative experimental results demonstrate that the DSDs observed by the DHI and fog monitor follow the Gamma distribution, and that the measured results of MVD are highly comparable. The analysis results of DSD reveal that the absence of the droplets with the size of 2–6 μm alters several patterns of microphysical parameters, and then the reliability of some observational data on cloud variations is diminished. As a result, the minimum size in DSD should be extended to 2 μm. The measured results of cloud microphysical parameters indicate a stronger positive correlation and increased robustness between microphysical parameters under the condition of high concentration aerosols. To further research the interaction mechanism of cloud microphysics and turbulence, an engineering prototype with higher sampling

frequency is required. Future research will examine the mechanism by which aerosol concentration influences the fluctuation of microphysical parameters.

**Author Contributions:** Conceptualization, P.G. and J.L.; methodology, P.G.; software, P.G. and J.W.; validation, P.G., J.W. and D.H.; formal analysis, P.G.; investigation, Y.G.; resources, Y.G.; data curation, P.G.; writing—original draft preparation, P.G.; writing—review and editing, P.G. and J.W.; visualization, P.G.; supervision, J.W.; project administration, Y.G.; funding acquisition, J.W. All authors have read and agreed to the published version of the manuscript.

**Funding:** This work was supported by the National Natural Science Foundation of China (41875034, 52127802 and 41975045) and scientific research project of Shanghai Meteorological Service (MS202004).

**Data Availability Statement:** Not applicable.

**Conflicts of Interest:** The authors declare no conflict of interest.

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
