# Peer review of "Observation on the Droplet Ranging from 2 to 16 μm in Cloud Droplet Size Distribution Based on Digital Holography"

_remotesensing, doi:10.3390/rs14102414_

Round 1

Reviewer 1 Report

I found the work interesting because it goes beyond the typical design criteria of holography and explores new features. I think the instrument is useful for measurements in environments where counting statistics with conventional instruments are not good. Nevertheless, I have some concerns about the methods and design criteria presented by the authors that should be clarified. The most important issue is the ability of the proposed system/code to detect particles as small as 2 µm at a distance of 6 mm from the image plane, which is about three times above the diffraction limit. How will this be achieved? How will this system perform in more challenging environments where noise (e.g., due to background light in the field) could affect the quality of the hologram? Detecting 2 µm particles at regular intervals on a grid is easy because it is known where to look for them. In a noisy environment, distinguishing these particles from noise beyond the diffraction limit is a major challenge. Another problem is the dependence between particle size and detection efficiency as a function of depth based on the diffraction limit. The detection efficiency of particles decreases with increasing depth, which must be taken into account. The authors can combine all the holograms obtained in a given experiment with statistically invariant droplet size distribution into one hologram; then a trend would be obtained showing that the detection efficiency for a particle is highest near the image and lower near the x-y edges and at high z, from which calibration curves can be derived. In addition, Fig. 1 should clearly indicate dimensions, exact details of optical components, shaded colors (what is dark green?), direction of sample flow (if any), etc. Camera and laser models should also be mentioned to make the system reproducible. In Fig. 7, I don't understand why 10 sample volumes are arranged one behind the other in z-direction and not side by side? What is new about the stitching technique, which is only mentioned in two places in the paper? When it comes to stitching multiple reconstructed holograms together to improve statistics, this is a widely used technique. When it comes to merging different hologram sections into a single coherent hologram based on the velocity of the sample flow, then this needs to be clearly explained. What is the detection efficiency of FM-120 as a function of particle size? Especially near 2µm?
Is the proposed instrument suitable for outdoor operation, such as on a building roof or mountain weather station? Please explain the limitations and possible applications. Further details on the reconstruction code, computation time, and sizing techniques should also be provided. If this is a real-time instrument, the authors have solved a major holographic problem. Then additional details about the code would be needed to understand how they achieved this.  With these changes, I believe the research is complete and useful to the field. 

Author Response

Thanks for reviewing our manuscript. Your suggestions and comments help us improve our current research.

Reviewer 2 Report

A new method was proposed in this study to measure cloud droplet size in lab, which is especially useful for very small cloud droplets. Such a method is based on the digital holography technique and seems to be appropriate and robust. Also, the manuscript is well organized and all the results are clearly presented. I believe this manuscript is almost ready for publication. The only somewhat negative consideration is that this study is not very suitable for the scope of this journal “Remote Sensing”, since it is an optical technique just used for in situ measurement aiming at cloud droplet size, rather than the way of remote sensing. Anyway, some suggests are as follows.

Minor Points

L2. In the title and other places, the statement “droplet with the size of 2 -16 um” is not very appropriate. It should be such as “droplet ranging from 2 to 16 um”.

L62-69. Statements herein are mostly unprofessional. For instance, “the trend of”. It is obviously not trend. Trend has a specific meaning. It may be “pattern”. Others like “change processes”, “opposite change trend”, “Under condition of the absence of”, should be simplified to be “In the absence of”, etc.

L251. It is very strange for the occurrence of “three days” herein and those “day” related statements later. It should have been introduced in Section 3. Actually, I believe the information of “day” is not relevant. It is just three experiments and each experiment continues 5 minutes. Readers need not to know the day, or the hour, that the experiment was conducted, unless some situations are related to the specific time.

Figure 12. It is not appropriate to connect those MVD-ED points as a green line, which should be remained as scattered points. Such a line makes no sense.

L400-406. Statements herein are not clear.

L439, “increased to 2 um”, should be “extended to 2 um”.

Author Response

(The authors gave the same response as above.)

Reviewer 3 Report

See attachment

Author Response

(The authors gave the same response as above.)

Round 2

Reviewer 1 Report

The authors have responded clearly enough to my comments. However, the authors' answers to questions 1 (diffraction limit etc) and 2 (noise handling methods) should also appear in the main manuscript (albeit in a modified form and in a concise manner) to make the work and methods understandable to all potential readers. 
